Three new species of subterranean amphipods (Pseudocrangonyctidae: Pseudocrangonyx) from limestone caves in South Korea

Lee Chi-Woo
Min Gi-Sik mingisik@inha.ac.kr
Department of Biological Sciences, Inha University , Incheon , South Korea
Glasby Christopher
Electronic publication date: 2021 Jan 27
Publication date: 2021
Volume: 9
Electronic Location ID: e10786
Received 2020 Oct 23; Accepted 2020 Dec 24
Copyright: ©2021 Lee and Min
Copyright year: 2021
Copyright holder: Lee and Min
License: This is an open access article distributed under the terms of the Creative Commons Attribution License, which permits unrestricted use, distribution, reproduction and adaptation in any medium and for any purpose provided that it is properly attributed. For attribution, the original author(s), title, publication source (PeerJ) and either DOI or URL of the article must be cited.
License URL: https://creativecommons.org/licenses/by/4.0/

Keywords: Crangonyctoidea, Identification key, Molecular analyses, Morphological variations, Stygobitic fauna

Funding: Nakdonggang National Institute of Biological Resources (NNIBR) Ministry of Environment (MOE) of the Republic of Korea NNIBR111-1301 NNIBR201901203 Inha University This study work was supported by a grant from the Nakdonggang National Institute of Biological Resources (NNIBR) funded by the Ministry of Environment (MOE) of the Republic of Korea (NNIBR 111-1301, NNIBR201901203), and by Inha University. The funders had no role in study design, data collection and analysis, decision to publish, or preparation of the manuscript

==============================
Pseudocrangonyx Akatsuka & Komai, 1922 is the most diverse group of subterranean amphipods in the groundwater communities of Far East Asia. In Korea, the diversity of the group has been underestimated due to the records of morphological variants of Pseudocrangonyx asiaticus Uéno, 1934. To estimate the species diversity, we analyzed the morphological characteristics and conducted molecular analyses of specimens collected from Korean caves that we treated as morphological variants of P. asiaticus. We described three new subterranean pseudocrangonyctid amphipod species, P. deureunensis sp. nov., P. kwangcheonseonensis sp. nov., and P. hwanseonensis sp. nov., from the groundwater of limestone caves in South Korea. Additionally, we determined sequences of the nuclear large subunit ribosomal RNA and the mitochondrial cytochrome c oxidase subunit I gene of the new species for molecular analyses. Molecular phylogenetic analyses revealed that the three new species formed a monophylum together with P. joolaei Lee et al., 2020 and P. wonkimi Lee, Tomikawa & Min, 2020, which are species that are endemic to Korean caves.

Introduction

Amphipods are the most diverse group of organisms in groundwater communities (Holsinger, 1994), and subterranean amphipods are even more notable from a biogeographic perspective because of their limited dispersal ability and restriction to groundwater aquifers (Holsinger, 1993). Most subterranean amphipods are troglobiont (stygobiont) and are generally characterized by morphological features such as appendage elongation and the loss of eyes and pigment (Holsinger, 1994; Väinölä et al., 2008). These characteristics result in the strikingly convergent morphology of these cave animals (Jones, Culver & Kane, 1992). Classifying subterranean organisms solely on their morphological characteristics leads to several taxonomic problems (Lefébure et al., 2006; Kornobis et al., 2011). Because subterranean and cave amphipod species are particularly difficult to morphologically identify, molecular analyses help in species delimitation (Lefébure et al., 2006; Trontelj et al., 2009; Hou & Li, 2010).

The stygobitic amphipod genus Pseudocrangonyx Akatsuka & Komai, 1922 is the most diverse taxon among the subterranean amphipod genera found in Far East Asia, i.e., the Korean Peninsula, the Japanese Archipelago, eastern China, and the Russian Far East (Sidorov & Holsinger, 2007; Tomikawa & Nakano, 2018). The first record of P. asiaticus Uéno, 1934 on the Korean Peninsula was from North Korea (Uéno, 1940). However, P. asiaticus’s type locality is on China’s Liaodong Peninsula (Uéno, 1934). Using identification techniques based on morphological characteristics, this species has been found to inhabit several caves in South Korea (Uéno, 1966; Holsinger, 1989). Uéno (1966) mentioned regional morphological variants of the Korean populations, but did not regard them as a distinct species. At that time, there were obvious limitations to correctly identifying subterranean amphipods based solely on morphological characteristics. Recent studies have used molecular analyses and morphological identification to show that the genus’ species diversity may be higher than previously believed (Tomikawa et al., 2016; Tomikawa & Nakano, 2018; Lee et al., 2020).

While conducting cave surveys on the Korean Peninsula, we collected Pseudocrangonyx specimens from two caves (Kwangcheonseon Cave and Hwanseon Cave) where Uéno (1966) reported finding one of the P. asiaticus Uéno, 1934 morphological variants. Additionally, specimens were also collected from Deureune Cave, where the genus Pseudocrangonyx had not been previously located. Based on the results of morphological examination of the amphipods, we described and illustrated them as three new Pseudocrangonyx species. Furthermore, we determined the nuclear large subunit ribosomal RNA (28S rRNA) gene and mitochondrial cytochrome c oxidase subunit I (COI) gene sequence data for molecular analyses of the three new species. Additionally, we provided a key to the Korean Pseudocrangonyx species.

Materials & Methods

Sample collection and morphological examination

Pseudocrangonyx specimens were collected from the groundwater of three Korean caves: Deureune Cave (Fig. 1A), Kwangcheonseon Cave (Fig. 1B), and Hwanseon Cave (Fig. 1C). We fixed and preserved the specimens in 99% ethanol. All specimen appendages were dissected in 80% ethanol and mounted with gum-chloral medium on glass slides under an Olympus SZX7 stereomicroscope (Tokyo, Japan). The specimens were examined using a Nikon Eclipse Ni light microscope (Tokyo, Japan) and were illustrated with the aid of a drawing tube. The body length from the tip of the rostrum to the base of the telson was measured along the dorsal curvature to the nearest 0.1 mm. The nomenclature of the setal patterns on the mandibular palp followed the method described by Stock (1974). The specimens examined in this study have been deposited in the collection at the Nakdonggang National Institute of Biological Resources, Korea (NNIBR).

Figure 1 The collection locality of the specimens examined in this study.

(A) Deureune cave; (B) Kwangcheonseon cave; (C) Hwanseon cave.

Table 1 Data used for molecular analyses.

Sequences marked with an asterisk were obtained for the first time in the present study.

Species	Voucher or isolate	Locality or Country
(Habitat)	GenBank No.	
			28S	COI	
Genus Pseudocrangonyx	
P. deureunensissp. nov.	NNIBRIV39838	Bonghwa, Korea
(Cave)	MW026427 *	MW026424 *	
NNIBRIV39835		MW026425 *	
NNIBRIV39839		MW026426 *	
P. kwangcheonseonensissp. nov.	NNIBRIV35120	Pyeongchang, Korea
(Cave)	MW026433 *	MW026430 *	
NNIBRIV39840		MW026431 *	
NNIBRIV39841		MW026432 *	
P. hwanseonensissp. nov.	NNIBRIV35118	Samcheok, Korea
(Cave)	MW026439 *	MW026436 *	
NNIBRIV39836		MW026437 *	
NNIBRIV39837		MW026438 *	
P. wonkimi	NNIBRIV35119	Hampyeong, Korea
(Cave)	MT316536	MT316534	
NNIBRIV36158		MT316535	
P. joolaei	NNIBRIV21629	Goesan, Korea
(Cave)	LC467007	LC467001	
NNIBRIV21630		LC467002	
P. daejeonensis	NNIBRIV1	Daejeon, Korea
(Interstitial water)	LC322136	LC322137	
P. akatsukai	KUZ Z1967 (G1277)	Yamaguchi, Japan	LC171506	LC171507	
P. komaii	KUZ Z1976 (G1297)	Yamaguchi, Japan	LC171541	LC171542	
P. gudariensis	NSMT-Cr 24605	Aomori, Japan	LC171498	LC171499	
P. yezonis	KUZ Z1970 (G1280)	Hokkaido, Japan	LC171518	LC171519	
P. uenoi	KUZ Z1964 (G405)	Shiga, Japan	LC171491	LC171492	
P. elegantulus	IZCAS I-A1602-2	China	KY436646	KY436647	
P. holsingeri		Russian Far East	KJ871679	KF153111	
P. korkishkoorum	B1	Russian Far East	KJ871678	KF153107	
P. korkishkoorum	N1	Russian Far East	KJ871676	KF153105	
P. tiunovi		Russian Far East	KJ871674	KF153110	
P. febras		Russian Far East		KF153114	
P. susanaensis		Russian Far East		KF153113	
P. sympatricus		Russian Far East		KF153112	
Outgroup Genus Crangonyx	
C. floridanus	G1322	Chiba, Japan	LC171549	LC171550	

Molecular analyses

We extracted genomic DNA from the muscles of the specimen appendages using the LaboPass Tissue Mini Kit (Cosmo GENETECH, Seoul, South Korea), according to the manufacturer’s instructions. We used the following primer sets for the PCR reaction used in this study: 28F and 28R for 28S rDNA (Hou, Fu & Li, 2007), and LCO1490 and HCO2198 for COI (Folmer et al., 1994). The sequences of 28S rDNA were aligned using MAFFT v. 7.388 L-INS-i (Katoh & Standley, 2013), and COI was aligned using Geneious 8.1.9 (Biomatters, Auckland, New Zealand). For phylogenetic analysis, these two alignments were combined. All data used in molecular analyses were provided, including the newly obtained sequences (Table 1). Pairwise comparisons of uncorrected p-distances for COI sequences were calculated using MEGA X (Kumar et al., 2018). Phylogenetic trees were constructed using maximum likelihood (ML) and Bayesian inference (BI). We performed ML analysis using RAxML v. 8.2.10 (Stamatakis, 2014) with the substitution model immediately set as GTRCAT after nonparametric bootstrapping was conducted with 1,000 replicates. The best fit-partitioning scheme for the ML analysis was identified with the Akaike information criterion using PartitionFinder v. 2.1.1 (Lanfear et al., 2017) with the “greedy” algorithm. BI and posterior probabilities were estimated using MrBayes v. 3.2.6 (Ronquist et al., 2012). Two independent runs of four Markov chains were conducted for 10 million generations, and the tree was sampled at every 100 generations. Parameter estimates and convergence were checked using Tracer v. 1.7.1 (Rambaut et al., 2018), and the first 50001 trees were discarded based on results.

Scanning electron microscopy

To carry out scanning electron microscope (SEM) imaging, we rinsed the specimen with TWEEN 20 (Model 036K00963; Sigma, St. Louis, MO, USA) to remove residual debris, and then dehydrated the sample with a graded ethanol series (30%, 50%, 70%, 80%, 95%, and 100% ethanol; 10 min each) and hexamethyldisilazane (Sigma, St. Louis, MO, USA) for 1 h. The dried sample was sputtered with platinum, and then observed using an SEM (Model Hitachi S-4300; Japan).

Terminology

Pseudocrangonyx asiaticus sensu stricto refers to the species that was originally described by Uéno (1934).

The electronic version of this article in Portable Document Format (PDF) will represent a published work according to the International Commission on Zoological Nomenclature (ICZN), and hence the new names contained in the electronic version are effectively published under that Code from the electronic edition alone. This published work and the nomenclatural acts it contains have been registered in ZooBank, the online registration system for the ICZN. The ZooBank LSIDs (Life Science Identifiers) can be resolved and the associated information viewed through any standard web browser by appending the LSID to the prefix http://zoobank.org/. The LSID for this publication is: [urn:lsid:zoobank.org:pub:A60F095A-2A50-4D87-876C-6D8E3D8539CE]. The online version of this work is archived and available from the following digital repositories: PeerJ, PubMed Central and CLOCKSS.

Results

Order Amphipoda Latreille, 1816	
Family Pseudocrangonyctidae Holsinger, 1989	
Genus PseudocrangonyxAkatsuka & Komai, 1922	
Pseudocrangonyx deureunensissp. nov.	
New Korean name: deu-reu-ne-dong-gul-yeop-sae-u	
(Figs. 2A, 3–8)	

Material examined. Holotype female (9.8 mm), NNIBRIV39838, collected from Deureune Cave (37°4.75′N, 128°59.36′E), Bonghwa-gun, Gyeongsangbuk-do, Korea, on 25 May 2018, by C. -W. Lee. Paratypes: 1 female (8.2 mm), NNIBRIV39835 (Fig. 2A); 1 male (7.1 mm), NNIBRIV39839, collection data same as that for the holotype.

Figure 2 Habitus of three new species.

(A) Paratype of Pseudocrangonyx deureunensis sp. nov., lateral view; (B) Holotype of Pseudocrangonyx kwangcheonseonensis sp. nov., lateral view; (C) Holotype of Pseudocrangonyx hwanseonensis sp. nov., lateral view.

Figure 3 Holotype of Pseudocrangonyx deureunensis sp. nov. (NNIBRIV39838).

(A) Head, lateral view; (B) Epimeral plates 1–3 and urosomites 1–3, lateral view; (C) Antenna 1, medial view; (D) Accessory flagellum of antenna 1, medial view; (E) Antenna 2, medial view; (F) Calceolus of antenna 2, medial view; (G) Upper lip, posterior view; (H) Left mandible, medial view; (I) Incisor, lacinia mobilis, and molar process of right mandible, medial view; (J) Incisor and lacinia mobilis of left mandible, medial view; (K) Lower lip, ventral view; (L) Maxilla 1, dorsal view.

Figure 4 Holotype of Pseudocrangonyx deureunensis sp. nov. (NNIBRIV39838).

(A) Maxilla 2, dorsal view; (B) Maxilliped, dorsal view; (C) Apical setae on inner plate of maxilliped, dorsal view; (D) Gnathopod 1, medial view; (E) Serrate setae on posterodistal corner of carpus of gnathopod 1, lateral view; (F) Palmar margin of propodus and dactylus of gnathopod 1, medial view; (G) Gnathopod 2, medial view; (H) Serrate setae on posterodistal corner of carpus of gnathopod 2, lateral view; (I) Palmar margin of propodus and dactylus of gnathopod 2, medial view.

Diagnosis. Antennal sinus with rounded angle; eyes absent; pereonites 1–6 with short dorsal setae; dorsal margin of urosomites 1–3 with setae; pereonites 2–4 each with sternal gill; antenna 1 being 0.51 times as long as body length; antenna 2 with calceoli in both sexes; mandible palp article 3 longer than article 2; maxilla 1 outer plate with 7 serrate teeth; maxilla 2 inner plate with oblique inner row of 6 setae; gnathopods 1 and 2, carpi with serrate setae on posterodistal corners in both sexes; palmar margins of propodi of gnathopods 1 and 2 with 11–15 and 14–18 robust setae, respectively; pleopod peduncles lacking marginal setae, inner margin of inner rami with bifid setae; uropod 1 inner ramus 0.9 times as long as peduncle, inner and outer margins of inner ramus with 3 and 2 robust setae, respectively, basal part of inner ramus with 3 slender setae, outer ramus with 2 marginal robust setae; uropod 2 inner ramus 1.2 times as long as peduncle, outer ramus with 2 marginal robust setae; uropod 3 terminal article longer than adjacent robust setae; telson each lobe with 2 apical robust setae and 1 penicillate seta.

Figure 5 Holotype of Pseudocrangonyx deureunensis sp. nov. (NNIBRIV39838).

(A) Pereopod 3, medial view; (B) Dactylus of pereopod 3, medial view; (C) Pereopod 4, medial view; (D) Dactylus of pereopod 4, medial view; (E) Pereopod 5, medial view; (F) Dactylus of pereopod 5, medial view; (G) Pereopod 6, medial view; (H) Dactylus of pereopod 6, medial view; (I) Pereopod 7, medial view; (J) Dactylus of pereopod 7, medial view.

Figure 6 Holotype of Pseudocrangonyx deureunensis sp. nov. (NNIBRIV39838).

(A) Sternal gills on pereonites 2–4, lateral view; (B) Pleopod 1, lateral view; (C) Retinacula on peduncle of pleopod 1, lateral view; (D) Pleopod 2, lateral view; (E) Retinacula on peduncle of pleopod 2, lateral view; (F) Pleopod 3, lateral view; (G) Retinacula on peduncle of pleopod 3, lateral view; (H) Uropod 1, dorsal view; (I) Uropod 2, dorsal view; (J) Uropod 3, dorsal view; (K) Telson, dorsal view.

Figure 7 Paratype of Pseudocrangonyx deureunensis sp. nov. (NNIBRIV39839).

(A) Antenna 1, medial view; (B) Accessory flagellum of antenna 1, medial view; (C) Antenna 2, medial view; (D) Calceolus of antenna 2, medial view; (E) Gnathopod 1, medial view; (F) Palmar margin of propodus and dactylus of gnathopod 1, medial view; (G) Gnathopod 2, medial view; (H) Palmar margin of propodus and dactylus of gnathopod 2, medial view.

Description. Female holotype (NNIBRIV39838). Head (Fig. 3A) with short dorsal seta; rostrum short; lateral cephalic lobe rounded; antennal sinus with rounded angle; eyes absent. Pereonites 1–6 with short dorsal setae; dorsal margin of pereonite 7 with long setae. Dorsal margins of pleonites 1–3 with long setae (Fig. 3B). Posterior margin of epimeral plate 1 with 6 setae, posteroventral corner with seta; ventral and posterior margins of plate 2 with 3 and 4 setae, respectively, posteroventral corner with seta; ventral and posterior margins of plate 3 with 3 setae, respectively, posteroventral corner subquadrate with seta (Fig. 3B). Dorsal margin of urosomites 1–3 with setae. Anteroventral corner of urosomite 1 with seta, posteroventral corner of urosomite 3 with setae (Fig. 3B).

Antenna 1 (Fig. 3C) 0.51 times as long as body length, peduncular articles 1–3 in length ratio of 1.0: 0.7: 0.4; accessory flagellum (Fig. 3D) 2-articulate, more than shorter primary flagellar article 1, terminal article with 3 setae and 1 aesthetasc; primary flagellum 18-articulate, 1 aesthetasc on some articles. Antenna 2 (Figs. 3E, 3F) 0.59 times as long as antenna 1; peduncular article 5 with 2 calceoli; flagellum 0.65 times as long as peduncular articles 4 and 5 combined, consisting of 9 articles, first 5 each with calceolus.

Figure 8 Paratype of Pseudocrangonyx deureunensis sp. nov. (NNIBRIV39839).

(A) Uropod 1, dorsal view; (B) Uropod 2, ventral view; (C) Distal robust seta on inner ramus of uropod 2, ventral view; (D) Uropod 3, dorsal view; (E) Terminal article of uropod 3, dorsal view; (F) Telson, dorsal view.

Upper lip (Fig. 3G) with rounded anterior margin, with fine setae. Mandibles (Figs. 3H–3J) with left and right incisors with 5-dentate, respectively; left lacinia mobilis 5-dentate, right lacinia bifid, with many teeth; molar process triturative; accessory setal rows of left and right mandibles each with 6-pectinate setae, respectively; palp 3-articulate, article 3 with 5 A-, 15 D-, and 2 E-setae. Lower lip (Fig. 3K) with broad outer lobes with fine setae, mandibular process of outer lobe rounded apically; inner lobes indistinct. Maxilla 1 (Fig. 3L) with inner and outer plates, and palp; inner plate subovate with 5 plumose setae; outer plate subrectangular with 7 serrate teeth apically; palp 2-articulate, longer than outer plate, article 2 with weakly plumose robust seta apically, and 3 apical and 4 subapical robust setae. Maxilla 2 (Fig. 4A) with oblique inner row of 6 setae on inner plate. Maxilliped (Figs. 4B, 4C) with inner and outer plates, and palp; inner plate subrectangular with four apical robust setae; outer plate suboval with apical and subapical four robust setae, and some medial setae; palp 4-articulate, medial margin of article 2 lined with setae, article 4 with nail.

Gnathopod 1 (Figs. 4D, 4E) with subquadrate coxa, bearing seta on anterior dorsal margin and anterodistal corner, width 1.8 times as long as depth; basis thick and short, anterior margin bare, submargin with setae, posterior margin with 16 long setae; posterodistal corner of carpus with 2 serrate robust setae; propodus stout, subtriangular, palmar margin with 15 robust setae in two rows, some distally notched; posterior margin of dactylus dentate (Fig. 4F). Gnathopod 2 (Figs. 4G, 4H) with rounded coxa, with setae on its anterior to ventral margins, width 1.4 times as long as depth; basis slender with anterior margin bare, posterior margin with 8 long setae; posterodistal corner of carpus with 3 serrate robust setae; propodus more slender than that of gnathopod 1, palmar margin with 18 robust setae in 2 rows, some distally notched; posterior margin of dactylus dentate (Fig. 4I). Pereopod 3 (Fig. 5A) with subquadrate coxa bearing setae on anterodistal corner to ventral margins, width 1.5 times as long as depth; anterior and posterior margins of basis with setae; merus, carpus, and propodus in length ratio of 1.0: 0.9: 0.9; posterior margin of dactylus with two setae (Fig. 5B). Pereopod 4 (Fig. 5C) with subquadrate coxa bearing setae on anterior margin, anterodistal corner and ventral margin, width 1.6 times as long as depth; anterior and posterior margins of basis with setae; merus, carpus, and propodus in length ratio of 1.0: 0.9: 1.0; posterior margin of dactylus with 2 setae (Fig. 5D). Pereopod 5 (Fig. 5E) with bilobed coxa bearing setae on anterior and posterior lobes; anterior and posterior margins of basis with setae; merus, carpus, and propodus in length ratio of 1.0: 1.0: 1.1; anterior margin of dactylus with 2 setae (Fig. 5F). Pereopod 6 (Fig. 5G) anterior coxa broken; bearing setae on posterior lobe; anterior and posterior margins of basis with setae; merus, carpus, and propodus in length ratio of 1.0: 1.0: 1.1; anterior margin of dactylus with 2 setae (Fig. 5H). Pereopod 7 (Fig. 5I) with subtriangular coxa, ventral margin weakly concave, with seta on ventral margin and posterodistal corner; anterior and posterior margins of basis with setae; merus, carpus, and propodus in length ratio of 1.0: 1.2: 1.2; anterior margin of dactylus with 2 setae (Fig. 5J).

Sternal gills (Fig. 6A) on ventral surfaces of pereonites 2–4, respectively.

Coxal gills (Figs. 4G, 5A, 5C, 5E, 5G) on gnathopod 2 and pereopods 3–6.

Brood plates (Figs. 4G, 5A, 5C, 5E) slender with numerous setae, on gnathopod 2 and pereopods 3–5.

Peduncles of pleopods 1–3 (Figs. 6B, 6D, 6F) lacking marginal setae, outerdistal corners with 2 setae, respectively. Pleopods 1–3 with paired retinacula (Figs. 6C, 6E, 6G), inner ramus inner basal margin with 2, 2, and 1 bifid seta (clothes-pin seta), respectively; inner ramus of pleopods 1–3 10-, 9-, and 9-articulate, respectively; outer ramus of pleopods 1–3 11-, 10-, and 9-articulate, respectively.

Uropod 1 (Fig. 6H) with basofacial seta on peduncle; inner ramus 0.9 times as long as peduncle, inner and outer margins with 3 and 2 robust setae, respectively, basal part with 3 slender setae; outer ramus 0.7 times as long as inner, with 2 outer margin robust setae, inner margin bare. Uropod 2 (Fig. 6I) with inner ramus 1.2 times as long as peduncle, outer margin and marginal with 2 robust setae, respectively; outer ramus 0.7 times as long as inner ramus, inner margin bare and outer margin with 2 robust setae. Uropod 3 (Fig. 6J) with peduncle 0.24 times as long as outer ramus; inner ramus absent; outer ramus 2-articulate, proximal article with robust setae, terminal article 0.2 times as long as proximal article, with 3 distal setae.

Telson (Fig. 6K) laterally straight, length 1.26 times as long as wide, cleft for 36.5% of length, each telson lobe with 2 lateral penicillate setae, apical with 2 robust setae and penicillate seta.

Male paratype (NNIBRIV39839). Antenna 1 (Figs. 7A, 7B) 0.64 times as long as body length, primary flagellum 18-articulate, 1 aesthetasc on some articles. Antenna 2 (Figs. 7C, 7D) 0.6 times as long as antenna 1; flagellum 0.61 times as long as peduncular articles 4 and 5 combined, consisting of 8 articles, articles 1–2 with calceolus.

Gnathopod 1 (Fig. 7E) carpus with serrate seta on posterodistal corner; palmar margin of propodus with 11 robust setae in 2 rows, some distally notched (Fig. 7F). Gnathopod 2 (Fig. 7G) carpus with 2 serrate setae on posterodistal corner; palmar margin of propodus with 14 robust setae in 2 rows, some distally notched (Fig. 7H).

Uropod 1 (Fig. 8A) with 2 basofacial setae on peduncle; inner ramus 0.76 times as long as peduncle; outer margin and marginal with 3 and 2 robust setae, respectively, basal part with 2 slender setae; outer ramus with 2 outer margin robust setae. Uropod 2 (Figs. 8B, 8C) with peduncle 0.82 times as long as inner ramus; inner ramus 1.4 times as long as outer ramus, distal part with 2 serrate, 4 simple robust setae. Uropod 3 (Figs. 8D, 8E) with outer ramus terminal article 0.2 times as long as proximal article.

Telson (Fig. 8F) length 1.2 times as long as wide, cleft for 39.1% of length.

Distribution. Known only from the type locality.

Etymology. The specific name is an adjective derived from the name of the cave where the new species was found.

Remarks. Pseudocrangonyx deureunensis sp. nov. is morphologically most similar to P. joolaei Lee et al., 2020 in having (1) eyes completely absent, (2) pereonites 1–6 with short dorsal setae, (3) ventral surface of pereonites 2–4 has sternal gills, (4) antenna 2 with calceoli in both sexes, (5) maxilla 1 outer plate with 7 serrate teeth, and 6) inner rami of pleopods with bifid setae on inner margin. The new species can be clearly distinguished from P. joolaei by the following characters (features of P. joolaei in parentheses): (1) urosomite 3 with (without) dorsal setae, (2) antenna 1 longer (shorter) than half the body length, (3) carpi of male gnathopods 1 and 2 with 1–2 (with 3) serrate robust setae on posterodistal corner, (4) uropod 3 terminal article longer (shorter) than adjacent robust setae, and (5) telson with 2 (with 4) apical robust setae.

Pseudocrangonyx kwangcheonseonensissp. nov.	
[New Korean name: kwang-cheon-seon-dong-gul-yeop-sae-u]	
(Figs. 2B, 9–14)	

Pseudocrangonyx asiaticus. —Uéno, 1966: 506–518 (in part), Figs. 2–4, 5A–5K.

Figure 9 Holotype of Pseudocrangonyx kwangcheonseonensis sp. nov. (NNIBRIV35120).

(A) Antenna 1, medial view; (B) Accessory flagellum of antenna 1, medial view; (C) Antenna 2, lateral view; (D) Calceolus of antenna 2, lateral view; (E) Upper lip, posterior view; (F) Right mandible, medial view; (G) Incisor, lacinia mobilis, and molar process of right mandible, medial view; (H) Incisor, lacinia mobilis, and molar process of left mandible, medial view; (I) Lower lip, dorsal view; (J) Maxilla 1, dorsal view; (K) Apical robust setae on outer plate of maxilla 1, dorsal view; (L) Maxilla 2, dorsal view.

Figure 10 Holotype of Pseudocrangonyx kwangcheonseonensis sp. nov. (NNIBRIV35120).

(A) Maxilliped, dorsal view; (B) Gnathopod 1, medial view; (C) Serrate setae on posterodistal corner of carpus of gnathopod 1, lateral view; (D) Palmar margin of propodus and dactylus of gnathopod 1, medial view; (E) Gnathopod 2, medial view; (F) Serrate setae on posterodistal corner of carpus of gnathopod 2, lateral view; (G) Palmar margin of propodus and dactylus of gnathopod 2, medial view.

Figure 11 Holotype of Pseudocrangonyx kwangcheonseonensis sp. nov. (NNIBRIV35120).

(A) Pereopod 3, medial view; (B) Dactylus of pereopod 3, medial view; (C) Pereopod 4, medial view; (D) Dactylus of pereopod 4, medial view; (E) Pereopod 5, medial view; (F) Dactylus of pereopod 5, medial view; (G) Pereopod 6, medial view; (H) Pereopod 7, medial view; (I) Dactylus of pereopod 7, medial view.

Figure 12 Holotype of Pseudocrangonyx kwangcheonseonensis sp. nov. (NNIBRIV35120).

(A) Sternal gills on pereonites 2–4, lateral view; (B) Pleopod 1, lateral view; (C) Retinacula on peduncle of pleopod 1, lateral view; (D) Pleopod 2, lateral view; (E) Retinacula on peduncle of pleopod 2, lateral view; (F) Pleopod 3, lateral view; (G) Retinacula on peduncle of pleopod 3, lateral view; (H) Uropod 1, dorsal view; (I) Uropod 2, dorsal view; (J) Uropod 3, dorsal view; (K) Terminal article of uropod 3, dorsal view; (L) Telson, ventral view.

Material examined. Holotype female (10.6 mm), NNIBRIV35120, collected from Kwangcheonseon Cave (37°31.11′N, 128°27.05′E), Pyeongchang-gun, Gangwon-do, Korea, on 28 February 2017, by Y. G. Choi. Paratypes: 1 male (7.8 mm), NNIBRIV39840; 1 male (7.1 mm), NNIBRIV39841, collection data same as that for the holotype.

Figure 13 Paratype of Pseudocrangonyx kwangcheonseonensis sp. nov. (NNIBRIV39840).

(A) Antenna 1, lateral view; (B) Accessory flagellum of antenna 1, lateral view; (C) Antenna 2, medial view; (D) Gnathopod 1, medial view; (E) Serrate setae on posterodistal corner of carpus of gnathopod 1, lateral view; (F) Palmar margin of propodus and dactylus of gnathopod 1, medial view; (G) Gnathopod 2, medial view; (H) Serrate setae on posterodistal corner of carpus of gnathopod 2, lateral view; (I) Palmar margin of propodus and dactylus of gnathopod 2, medial view.

Figure 14 Paratype of Pseudocrangonyx kwangcheonseonensis sp. nov. (NNIBRIV39840).

(A) Uropod 1, dorsal view; (B) Uropod 2, dorsal view; (C) Uropod 3, dorsal view; (D) Terminal article of uropod 3, dorsal view; (E) Telson, dorsal view.

Diagnosis. Female larger than male; antennal sinus with rounded angle; eyes absent; pereonites 1–7 with dorsal setae; dorsal margin of urosomite 3 lacking setae; pereonites 2–4 each with sternal gill; antenna 1 longer than half body length; antenna 2 with calceoli in both sexes; mandible palp article 3 longer than article 2; maxilla 1 inner plate with 8 plumose setae; gnathopods 1 and 2, carpi with serrate setae on posterodistal corners in both sexes; palmar margins of propodi of gnathopods 1 and 2 with 24–26 and 20–21 robust setae, respectively; pleopod peduncles lacking marginal setae, inner margin of inner rami with bifid setae; uropod 1, inner and outer margins of inner ramus with 4 and 3 robust setae, basal part of inner ramus with 3 slender setae, outer ramus with 3 outer marginal robust setae; uropod 3 terminal article shorter than adjacent robust setae; telson laterally concave and shallowly at the top.

Description. Female holotype (NNIBRIV35120). Head (Fig. 2B) with short dorsal setae; rostrum short; lateral cephalic lobe rounded; antennal sinus shallow with rounded angle; eyes absent. Pereonites 1–6 with short dorsal setae; dorsal margin of pereonite 7 with long setae. Dorsal margins of pleonites 1–3 with long setae (Fig. 2B). Posterior margin of epimeral plate 1 with 5 setae; ventral and posterior margins of plate 2 with 2 and 4 setae, respectively, posteroventral corner with seta; ventral and posterior margins of plate 3 with 3 setae, respectively, posteroventral corner with seta (Fig. 2B). Dorsal margin of urosomites 1–2 with setae, urosomite 3 lacking dorsal setae. Anteroventral corner of urosomite 1 with seta, posteroventral corner of urosomite 3 with setae (Fig. 2B).

Antenna 1 (Fig. 9A) 0.56 times as long as body length, peduncular articles 1–3 in length ratio of 1.0: 0.7: 0.4; accessory flagellum (Fig. 9B) 2-articulate, more than longer primary flagellar article 1, terminal article with 3 setae and 1 aesthetasc; primary flagellum 21-articulate, 1 aesthetasc on some articles. Antenna 2 (Figs. 9C, 9D) 0.64 times as long as antenna 1; peduncular article 5 with 4 calceoli; flagellum 0.52 times as long as peduncular articles 4 and 5 combined, consisting of 8 articles, first 6 each with calceolus.

Upper lip (Fig. 9E) with rounded anterior margin, with fine setae. Mandibles (Figs. 9F–9H) with left and right incisors with 6- and 5-dentate, respectively; left lacinia mobilis 5-dentate, right lacinia bifid, with many teeth; molar process triturative; accessory setal rows of left and right mandibles with 7 and 6 pectinate setae; palp 3-articulate, article 3 with 8 A-, 20 D-, and 5 E-setae. Lower lip (Fig. 9I) with broad outer lobes with fine setae, mandibular process of outer lobe rounded apically; inner lobes indistinct. Maxilla 1 (Figs. 9J, 9K) with inner and outer plates, and palp; inner plate subovate with 8 plumose setae; outer plate subrectangular with 7 serrate teeth apically; palp 2-articulate, longer than outer plate, article 2 with plumose robust seta apically. Maxilla 2 (Fig. 9L) with oblique inner row of 10 setae on inner plate. Maxilliped (Fig. 10A) with inner and outer plates, and palp; inner plate subrectangular with 4 apical robust setae; outer plate suboval with apical and subapical 6 robust setae, and some medial setae; palp 4-articulate, medial margin of article 2 lined with setae, article 4 with nail.

Gnathopod 1 (Figs. 10B, 10C) with subquadrate coxa, bearing seta on its anterior to ventral margins, width 1.7 times as long as depth; basis thick and short, anterior margin bare, posterior margin with 15 long setae; posterodistal corner of carpus with 2 serrate robust setae; propodus stout, subtriangular, palmar margin with 24 robust setae in 2 rows, some distally notched; posterior margin of dactylus dentate (Figs. 10D). Gnathopod 2 (Figs. 10E, 10F) with rounded coxa, with setae on its anterior to ventral margins, width 1.3 times as long as depth; basis slender with anterior margin bare, posterior margin with 16 long setae; posterodistal corner of carpus with 3 serrate robust setae; propodus more slender than that of gnathopod 1, palmar margin with 21 robust setae in 2 rows, some distally notched; posterior margin of dactylus dentate (Fig. 10G). Pereopod 3 (Fig. 11A) with subquadrate coxa bearing setae on anterior margin to posteroventral corner, width 1.4 times as long as depth; basis posterior margin with 17 long setae; merus, carpus, and propodus in length ratio of 1.0: 0.7: 0.7; posterior margin of dactylus with 2 setae (Fig. 11B). Pereopod 4 (Fig. 11C) with subquadrate coxa bearing setae on anterior margin to posteroventral corner, width 1.6 times as long as depth; basis posterior margin with 12 long setae; merus, carpus, and propodus in length ratio of 1.0: 0.8: 0.7; posterior margin of dactylus with seta (Fig. 11D). Pereopod 5 (Fig. 11E) with bilobed coxa bearing setae on anterior and posterior lobes; anterior and posterior margins of basis with setae; merus, carpus, and propodus in length ratio of 1.0: 1.0: 0.9; anterior margin of dactylus with 1 seta (Fig. 11F). Pereopod 6 (Fig. 11G) with weakly bilobed coxa bearing setae on anterior and posterior lobes; anterior and posterior margins of basis with setae; merus, carpus, and propodus in length ratio of 1.0: 1.0: 0.9. Pereopod 7 (Fig. 11H) anterior coxa broken, ventral margin weakly concave, with setae on ventral margin and posterodistal corner; anterior and posterior margins of basis with setae; merus, carpus, and propodus in length ratio of 1.0: 1.1: 1.0; anterior margin of dactylus with 2 setae (Fig. 11I).

Sternal gills (Fig. 12A) on ventral surfaces of pereonites 2–4, respectively.

Coxal gills (Figs. 10E, 11A, 11C, 11E, 11G) on gnathopod 2 and pereopods 3–6.

Brood plates (Figs. 10E, 11A, 11C, 11E) slender with numerous setae, on gnathopod 2 and pereopods 3–5.

Peduncles of pleopods 1–3 (Figs. 12B, 12D, 12F) lacking marginal setae, outerdistal corners with 2, 4, and 1 seta, respectively. Pleopods 1–3 with paired retinacula (Figs. 12C, 12E, 12G), inner ramus inner basal margin with 3, 2, and 2 bifid seta (clothes-pin seta), respectively; inner ramus of pleopods 1–3 11-, 9-, and 9-articulate, respectively; outer ramus of pleopods 1–3 13-, 13-, and 10-articulate, respectively.

Uropod 1 (Fig. 12H) with basofacial seta on peduncle; inner ramus 0.76 times as long as peduncle, inner and outer margins with 4 and 3 robust setae, respectively, basal part with 3 slender setae; outer ramus 0.6 times as long as inner, with 3 outer marginal robust setae, inner margin bare. Uropod 2 (Fig. 12I) with inner ramus 1.1 times as long as peduncle, outer margin and marginal with 3 and 2 robust setae, respectively; outer ramus 0.7 times as long as inner ramus, inner margin bare and outer margin with 2 robust setae. Uropod 3 (Figs. 12J, 12K) with peduncle 0.29 times as long as outer ramus; inner ramus absent; outer ramus 2-articulate, proximal article with robust setae, terminal article 0.07 times as long as proximal article, with 3 distal setae.

Telson (Fig. 12L) base laterally concave and shallowly at the top, length 1.33 times as long as wide, cleft for 40.2% of length, each telson lobe with lateral penicillate setae, apical with 3 robust setae and 1 seta.

Male paratype (NNIBRIV39840). Antenna 1 (Figs. 13A, 13B) 0.54 times as long as body length, primary flagellum 18-articulate, 1 aesthetasc on some articles. Antenna 2 (Fig. 13C) 0.6 times as long as antenna 1; flagellum 0.46 times as long as peduncular articles 4 and 5 combined, consisting of 7 articles, peduncular article 5 without calceoli and some flagellum with calceolus.

Gnathopod 1 (Figs. 13D, 13E) carpus with 2 serrate setae on posterodistal corner; palmar margin of propodus with 26 robust setae in 2 rows, some distally notched (Fig. 13F). Gnathopod 2 (Figs. 13G, 13H) carpus with 2 serrate setae on posterodistal corner; palmar margin of propodus with 20 robust setae in 2 rows, some distally notched (Fig. 13I).

Uropod 1 (Fig. 14A) with basofacial seta on peduncle; inner ramus 0.81 times as long as peduncle; inner margin bare and outer margin with 3 robust setae, basal part with 2 slender setae; outer ramus with 1 marginal robust seta. Uropod 2 (Fig. 14B) with peduncle and inner ramus ratio 1.0: 1.0; inner ramus 1.2 times as long as outer ramus, distal part with 5 serrate, 3 simple robust setae, 1 simple seta. Uropod 3 (Figs. 14C, 14D) with outer ramus terminal article 0.1 times as long as proximal article.

Telson (Fig. 14E) length 1.43 times as long as wide, cleft for 45.7% of length.

Distribution. Known only from the type locality.

Etymology. The specific name is an adjective derived from the name of the cave where the new species was found.

Remarks. Pseudocrangonyx kwangcheonseonensis sp. nov. is morphologically most similar to P. asiaticus Uéno, 1934 in having (1) eyes completely absent, (2) sternal gills present (3) accessory flagellum of antenna 1 being as long as first article of primary flagellum, (4) antenna 2 longer than half of antenna 1 length, and (5) carpi of gnathopods 1 and 2 with serrate robust setae on posterodistal corner. The new species can be clearly distinguished from P. asiaticus by the following characters (features of P. asiaticus in parentheses): (1) pereonites 1–7 with (without) short dorsal setae, (2) sternal gills present on pereonites 2–4 (pereonites 2–5), (3) maxilla 1 inner plate with 8 (with 4) plumose setae, (4) antenna 1 longer (shorter) than half of the body length, and (5) inner ramus of uropod 1 with 3 (without) outer marginal robust setae.

Pseudocrangonyx hwanseonensissp. nov.	
[New Korean name: Hwan-seon-dong-gul-yeop-sae-u]	
(Figs. 2C, 15–20)	

Pseudocrangonyx asiaticus. —Uéno, 1966: 506–518 (in part), Figs. 5O, 7E.

Figure 15 Holotype of Pseudocrangonyx hwanseonensis sp. nov. (NNIBRIV35118).

(A) Antenna 1, medial view; (B) Accessory flagellum of antenna 1, medial view; (C) Antenna 2, medial view; (D) Calceolus of antenna 2, medial view; (E) Upper lip, posterior view; (F) Left mandible, medial view; (G) Incisor, lacinia mobilis, and molar process of right mandible, medial view; (H) Lower lip, ventral view; (I) Maxilla 1, dorsal view; (J) Maxilla 2, dorsal view.

Material examined. Holotype female (7.5 mm), NNIBRIV35118, collected from Hwanseon Cave (37°19.52′N, 129°1.02′E), Samcheok-si, Gangwon-do, Korea, on 20 October 2018, by Y. G. Choi. Paratypes: 1 female (7.7 mm), NNIBRIV39836; 1 male (6.3 mm), NNIBRIV39837, collection data same as that for the holotype.

Diagnosis. Female larger than male; antennal sinus with rounded angle; eyes absent; pereonites 1–6 without short dorsal setae; dorsal margin of urosomite 3 lacking setae; pereonites 2–4 each with 1 pair of sternal gills; antenna 1 0.53 times as long as body length; antenna 2 with calceoli in both sexes; mandible palp article 3 longer than article 2; maxilla 1 inner plate with 4 plumose setae; maxilla 2 inner plate with oblique inner row of 6 setae; gnathopods 1 and 2, carpi with serrate setae on posterodistal corners in both sexes; palmar margins of propodi of gnathopods 1 and 2 with 13 and 13–15 robust setae, respectively; pleopod peduncles lacking marginal setae, inner margin of inner rami with bifid setae; uropod 1 inner ramus 0.86 times as long as peduncle, inner and outer margins of inner ramus with 3 and 1 robust setae, respectively, basal part of inner ramus with 3 slender setae, outer ramus with 2 marginal robust setae; uropod 2 inner ramus 1.2 times as long as peduncle; inner and outer margins of inner ramus with 2 robust setae, respectively, outer ramus with 2 outer marginal robust setae; uropod 3 terminal article 0.15 time as long as length of proximal article; telson length 1.31 time as long as width, cleft for 36.8%.

Description. Female holotype (NNIBRIV35118). Head (Fig. 2C) without dorsal setae; rostrum short; lateral cephalic lobe rounded; antennal sinus shallow with rounded angle; eyes absent. Pereonites 1–6 without short dorsal setae; dorsal margin of pereonite 7 with long setae. Dorsal margins of pleonites 1–3 with long setae (Fig. 2C). Ventral and posterior margins of epimeral plate 1 with 1 and 5 setae, respectively, posteroventral corner with 1 seta; ventral and posterior margins of plate 2 with 4 and 5 setae, respectively, posteroventral corner with 1 seta; ventral and posterior margins of plate 3 with 4 setae, respectively, posteroventral corner with 1 seta (Fig. 2C). Dorsal margin of urosomites 1–2 with setae, urosomite 3 lacking dorsal setae. Anteroventral corner of urosomite 1 with 1 seta, posteroventral corner of urosomite 3 with setae (Fig. 2C).

Antenna 1 (Fig. 15A) 0.53 times as long as body length, peduncular articles 1–3 in length ratio of 1.0: 0.7: 0.4; accessory flagellum (Fig. 15B) 2-articulate, more than shorter primary flagellar article 1, terminal article with 3 setae and 1 aesthetasc; primary flagellum 16-articulate, 1 aesthetasc on some articles. Antenna 2 (Figs. 15C, 15D) 0.58 times as long as antenna 1; peduncular article 5 with two calceoli; flagellum 0.53 times as long as peduncular articles 4 and 5 combined, consisting of seven articles, first 3 each with calceolus.

Upper lip (Fig. 15E) with rounded anterior margin, with fine setae. Mandibles (Figs. 15F, 15G) left and right incisors 5-dentate; left lacinia mobilis 5-dentate, right lacinia bifid, with many teeth; molar process triturative; accessory setal rows of left and right mandibles with 5 and 4 pectinate setae; palp 3-articulate, article 3 with 5 A-, 12 D-, and 4 E-setae. Lower lip (Fig. 15H) with broad outer lobes with fine setae, mandibular process of outer lobe rounded apically; inner lobes indistinct. Maxilla 1 (Fig. 15I) with inner and outer plates, and palp; inner plate subovate with four plumose setae; outer plate subrectangular with seven serrate teeth apically; palp 2-articulate, longer than outer plate, article 2 with plumose robust seta apically. Maxilla 2 (Fig. 15J) with slender outer plate; oblique inner row of six setae on inner plate. Maxilliped (Fig. 16A) with inner and outer plates, and palp; inner plate subrectangular with six apical robust setae; outer plate suboval with apical and subapical 3 robust setae, and some medial setae; palp 4-articulate, medial margin of article 2 lined with setae, article 4 with nail.

Figure 16 Holotype of Pseudocrangonyx hwanseonensis sp. nov. (NNIBRIV35118).

(A) Maxilliped, dorsal view; (B) Gnathopod 1, lateral view; (C) Serrate setae on posterodistal corner of carpus of gnathopod 1, lateral view; (D) Palmar margin of propodus and dactylus of gnathopod 1, lateral view; (E) Gnathopod 2, lateral view; (F) Serrate setae on posterodistal corner of carpus of gnathopod 2, lateral view; (G) Palmar margin of propodus and dactylus of gnathopod 2, lateral view.

Figure 17 Holotype of Pseudocrangonyx hwanseonensis sp. nov. (NNIBRIV35118).

(A) Pereopod 3, lateral view; (B) Dactylus of pereopod 3, lateral view; (C) Pereopod 4, lateral view; (D) Dactylus of pereopod 4, lateral view; (E) Pereopod 5, lateral view; (F) Dactylus of pereopod 5, lateral view; (G) Pereopod 6, lateral view; (H) Dactylus of pereopod 6, lateral view; (I) Pereopod 7, lateral view; (J) Dactylus of pereopod 7, lateral view.

Figure 18 Holotype of Pseudocrangonyx hwanseonensis sp. nov. (NNIBRIV35118).

(A) Sternal gills on pereonites 2–4, lateral view; (B) Pleopod 1, lateral view; (C) Retinacula on peduncle of pleopod 1, lateral view; (D) Pleopod 2, lateral view; (E) Retinacula on peduncle of pleopod 2, lateral view; (F) Pleopod 3, lateral view; (G) Retinacula on peduncle of pleopod 3, lateral view; (H) Uropod 1, dorsal view; (I) Uropod 2, ventral view; (J) Uropod 3, dorsal view; (K) Terminal article of uropod 3, dorsal view; (L) Telson, dorsal view.

Figure 19 Paratype of Pseudocrangonyx hwanseonensis sp. nov. (NNIBRIV39837).

(A) Antenna 1, medial view; (B) Accessory flagellum of antenna 1, medial view; (C) Antenna 2, medial view; (D) Calceolus of antenna 2, medial view; (E) Gnathopod 1, lateral view; (F) Serrate setae on posterodistal corner of carpus of gnathopod 1, lateral view; (G) Palmar margin of propodus and dactylus of gnathopod 1, lateral view; (H) Gnathopod 2, lateral view; (I) Serrate setae on posterodistal corner of carpus of gnathopod 2, lateral view; (J) Palmar margin of propodus and dactylus of gnathopod 2, lateral view.

Gnathopod 1 (Figs. 16B, 16C) with subquadrate coxa, bearing setae on anterodistal corner to ventral margin, width 1.6 times as long as depth; basis thick and short, anterior margin with one seta and some medial setae, posterior margin with 10 long setae; posterodistal corner of carpus with two serrate robust setae; propodus stout, subtriangular, palmar margin with 13 robust setae in two rows, some distally notched; posterior margin of dactylus dentate (Fig. 16D). Gnathopod 2 (Figs. 16E, 16F) with subrounded coxa, with setae on its anterior to ventral corners, width 1.3 times as long as depth; basis slender with anterior and posterior margin with 1 seta and 11 long setae, respectively; posterodistal corner of carpus with three serrate robust setae; propodus more slender than that of gnathopod 1, palmar margin with 15 robust setae in two rows, some distally notched; posterior margin of dactylus dentate (Fig. 16G). Pereopod 3 (Fig. 17A) with subquadrate coxa bearing setae on anterior margin to posteroventral corner, width 1.4 times as long as depth; anterior and posterior margins of basis with setae; merus, carpus, and propodus in length ratio of 1.0: 0.8: 0.8; posterior margin of dactylus with two setae (Fig. 17B). Pereopod 4 (Fig. 17C) with subquadrate coxa bearing setae on anterodistal to posteroventral corners, width 1.4 times as long as depth; basis posterior margin with 9 long setae; merus, carpus, and propodus in length ratio of 1.0: 0.9: 0.8; posterior margin of dactylus with two setae (Fig. 17D). Pereopod 5 (Fig. 17E) with bilobed coxa bearing setae on anterior and posterior lobes; anterior and posterior margins of basis with setae; merus, carpus, and propodus in length ratio of 1.0: 0.9: 1.0; anterior margin of dactylus with 2 setae (Fig. 17F). Pereopod 6 (Fig. 17G) with weakly bilobed coxa bearing setae on anterior and posterior lobes; anterior and posterior margins of basis with setae; merus, carpus, and propodus in length ratio of 1.0: 0.9: 1.0; anterior margin of dactylus with 2 setae (Fig. 17H). Pereopod 7 (Fig. 17I) with subtriangular coxa, ventral margin weakly concave, with setae on ventral margin and posterodistal corner; anterior and posterior margins of basis with setae; merus, carpus, and propodus in length ratio of 1.0: 1.0: 1.1; anterior margin of dactylus with 2 setae (Fig. 17J).

Figure 20 Paratype of Pseudocrangonyx hwanseonensis sp. nov. (NNIBRIV39837).

(A) Uropod 1, dorsal view; (B) Uropod 2, dorsal view; (C) Uropod 3, dorsal view; (D) Terminal article of uropod 3, dorsal view; (E) Telson, dorsal view.

Sternal gills (Fig. 18A) on ventral surfaces of pereonites 2–4, paired.

Coxal gills (Figs. 16E, 17A, 17C, 17E, 17G) on gnathopod 2 and pereopods 3–6.

Brood plates (Figs. 16E, 17A, 17C, 17E) slender with numerous setae, on gnathopod 2 and pereopods 3–5.

Peduncles of pleopods 1–3 (Figs. 18B, 18D, 18F) lacking marginal setae, outerdistal corners with 2, 2, and 1 setae, respectively. Pleopods 1–3 with paired retinacula (Figs. 18C, 18E, 18G), inner ramus inner basal margin with 2, 2, and 1 bifid seta (clothes-pin seta), respectively; inner ramus of pleopods 1–3 9-, 8-, and 7-articulate, respectively; outer ramus of pleopods 1–3 10-, 9-, and 8-articulate, respectively.

Uropod 1 (Fig. 18H) with basofacial seta on peduncle; inner ramus 0.86 times as long as peduncle, inner and outer margins with 3 and 1 robust setae, respectively, basal part with 3 slender setae; outer ramus 0.76 times as long as inner, with 2 outer marginal robust setae, inner margin bare. Uropod 2 (Fig. 18I) with inner ramus 1.2 times as long as peduncle, outer margin and marginal with 2 robust setae, respectively; outer ramus 0.73 times as long as inner ramus, inner margin bare and outer margin with 2 robust setae. Uropod 3 (Figs. 18J, 18K) with peduncle 0.29 times as long as outer ramus; inner ramus absent; outer ramus 2-articulate, proximal article with robust setae, terminal article 0.15 times as long as proximal article, with 4 distal setae.

Telson (Fig. 18L) length 1.31 times as long as wide, cleft for 36.8% of length, each telson lobe with 2 lateral penicillate setae, apical robust setae and one short penicillate seta.

Male paratype (NNIBRIV39837). Antenna 1 (Figs. 19A, 19B) 0.53 times as long as body length, primary flagellum 14-articulate, 1 aesthetasc on some articles. Antenna 2 (Figs. 19C, 19D) 0.63 times as long as antenna 1; flagellum 0.58 times as long as peduncular articles 4 and 5 combined, consisting of 7 articles, first 2 each with calceolus.

Gnathopod 1 (Figs. 19E, 19F) carpus with two serrate setae on posterodistal corner; palmar margin of propodus with 13 robust setae in two rows, some distally notched (Fig. 19G). Gnathopod 2 (Figs. 19H, 19I) carpus with three serrate setae on posterodistal corner; palmar margin of propodus with 13 robust setae in two rows, some distally notched (Fig. 19J).

Uropod 1 (Fig. 20A) with basofacial seta on peduncle; inner ramus 0.79 times as long as peduncle; inner and outer margins with 3 and 1 robust setae, respectively, basal part with 3 slender setae; outer ramus with 2 margin robust setae. Uropod 2 (Fig. 20B) with peduncle 0.90 times as long as inner ramus; inner ramus 1.3 times as long as outer ramus, distal part with 6 serrate, 2 simple robust setae. Uropod 3 (Figs. 20C, 20D) with outer ramus terminal article 0.22 times as long as proximal article.

Telson (Fig. 20E) length 1.25 times as long as wide, cleft for 40.0% of length.

Distribution. Known only from the type locality.

Etymology. The specific name is an adjective derived from the name of the cave where the new species was found.

Remarks. Pseudocrangonyx hwanseonensis sp. nov. is morphologically similar to P. asiaticus Uéno, 1934 in having (1) eyes completely absent, (2) pereonites 1–6 without short dorsal setae, (3) urosomite 1 with ventral robust seta, (4) maxilla 1 inner plate with 4 plumose setae, (5) antenna 2 longer than half of antenna 1 length, and (6) carpi of gnathopods 1 and 2 with serrate robust setae on posterodistal corner. The new species can be clearly distinguished from P. asiaticus by the following characters (features of P. asiaticus in parentheses): (1) sternal gills of 1 pair (single) present on each pereonites 2–4 (pereonites 2–5), (2) maxilla 1 outer plate with 7 (with 5) serrate teeth, (3) antenna 1 longer (shorter) than as long as body length half, and (4) uropod 3 terminal article shorter (longer) than adjacent robust setae.

Key to the species of Korean Pseudocrangonyx

1	Sternal gills absent .............................................................................................................. 2	
–	Sternal gills present ............................................................................................................. 4	
2	Female body size larger than 6.0 mm ................................................................................. 3	
–	Female body size smaller than 6.0 mm ................................ P. daejeonensisLee et al., 2018	
3	Uropod 3 terminal article longer than adjacent robust setae ...... P. minutusJung et al., 2020	
	− Uropod 3 terminal article shorter than adjacent robust setae ...... P. wonkimiLee, Tomikawa & Min, 2020	
4	Carpus of gnathopod with serrate robust setae on posterodistal corner ......................... 5	
–	Carpus of gnathopod without serrate robust setae on posterodistal corner .................... ......................................................................................................... P. coreanusUéno, 1966	
5	Sternal gills on pereonites 2 to 4 ......................................................................................... 6	
–	Sternal gills on pereonites 2 to 5 ......................................................................................... 9	
6	Sternal gills total number 3 ................................................................................................. 7	
–	Sternal gills total number 6 .......................................................... P. hwanseonensissp. nov.	
7	Urosomite 3 without dorsal setae ........................................................................................ 8	
–	Urosomite 3 with dorsal setae ........................................................ P. deureunensis sp. nov.	
8	Accessory flagellum of antenna 1 exceeding first article of primary flagellum ..................... .........................................................................P. kwangcheonseonensissp. nov.	
–	Accessory flagellum of antenna 1 not exceeding first article of primary flagellum ............... .......................................................................................................... P. joolaeiLee et al., 2020	
9	Maxilla 1 inner plate with less than 7 plumose setae ........................................................ 10	
–	Maxilla 1 inner plate with 7 plumose setae ............................... P. villosusJung et al., 2020	
10	Telson cleft less than 40% for length ................................................................................ 11	
–	Telson cleft more than 40% for length .............................................................................. 12	
11	Sternal gills on pereonites 2 to 5 (1+1+1+1) .................................... P. asiaticusUéno, 1934	
–	Sternal gills on pereonites 2 to 5 (1+1+0+1) .............................. P. crassusJung et al., 2020	
12	Uropod 2 outer ramus with 2 inner marginal robust setae ...... P. gracilipesJung et al., 2020	
–	Uropod 2 outer ramus without inner marginal robust setae ..... P. concavusJung et al., 2020	

Molecular Analyses. The uncorrected COI p-distance among the species of the genus Pseudocrangonyx in Korean caves is shown in Table 2; this divergence was calculated based on the 657 aligned positions from the data set. The range of interspecific variation was 11.7–17.0%. However, the maximum intraspecific variation was 0.2% within each species. In the phylogenetic analyses (Fig. 21), the topologies of the BI and ML trees were almost identical. Results of the present analyses showed that the species of the genus Pseudocrangonyx, inhabiting individual caves, were distinct new species.

Table 2 Intra- and interspecific variation calculated from COI of Korean cave Pseudocrangonyx.

Species name	Intraspecific (%)	Interspecific (%)	
		1	2	3	4	5	
P. deureunensis sp. nov.	0.2	–					
P. kwangcheonseonensis sp. nov.	0.2	13.2–13.4	–				
P. hwanseonensis sp. nov.	–	13.2–13.4	12.5–12.6	–			
P. wonkimi	–	16.9–17.0	13.9–14.0	14.6	–		
P. joolaei	0.2	15.2–15.4	14.8–15.1	11.7–11.9	14.9–15.1	–	

Figure 21 Maximum likelihood and Bayesian inference analyses based on nuclear 28S rRNA and mitochondrial COI sequences.

Numbers on nodes represent bootstrap values for maximum likelihood and Bayesian posterior probabilities.

Discussion

The three new species described in this paper are similar to P. asiaticus Uéno, 1934 in morphology, and they share the following characteristics: relatively large body size (about 8.0–10.0 mm), completely absent eyes, presence of basal setae on urosomite 1, present sternal gills, and carpi of gnathopods 1 and 2 with serrate robust setae on the posterodistal corner. However, the three new species have following characteristics that distinguished them as distinct new species: (1) P. deureunensis sp. nov., urosomite 3 with dorsal setae; (2) P. kwangcheonseonensis sp. nov., maxilla 1 inner plate with 8 plumose setae, telson base laterally concave, and shallow at the top; and (3) P. hwanseonensis sp. nov., sternal gills of 1 pair present on each pereonites 2–4. Furthermore, the COI genetic distance among the three species showed significant differentiation (12.5–13.4%) that was sufficient to designate the species as distinct , which was confirmed by a previous study (12–20%) on the genus Pseudocrangonyx (Zhao & Hou, 2017).

Most geographically separated subterranean species are likely to be independent in origin due to their poor dispersal and small ranges (Trontelj et al., 2009; Trontelj, Blejec & Fišer, 2012). Likewise, our molecular phylogenetic analyses revealed that the species within the genus Pseudocrangonyx endemic to the Korean Peninsula caves formed a monophyletic clade (Fig. 21), suggesting that the genus Pseudocrangonyx inhabits groundwater environments where dispersal is limited, such as a cave, and that they may have an independent origin in each habitat. A previous study (Lee et al., 2020) found that P. joolaei Lee et al., 2020 and P. akatsukai Tomikawa & Nakano, 2018 formed a clade. However, our results showed that P. joolaei has a closer relationship with the Korean cave Pseudocrangonyx species. Additionally, our phylogenetic analyses showed that all Korean cave Pseudocrangonyx species were a same clade, but the Japanese P. akatsukai was in a different clade. This means that rather than forming a clade with the Japanese species, the Korean cave Pseudocrangonyx species may share a single lineage with P. asiaticus sensu stricto, which is geographically adjacent and morphologically similar. Unfortunately, we could not obtain molecular data for P. asiaticus Uéno, 1934, and it is unclear whether P. asiaticus sensu stricto inhabit the Korean Peninsula. Additional molecular data for P. asiaticus should be examined in order to confirm the presence of P. asiaticus sensu stricto and explore the true species diversity of Pseudocrangonyx amphipods inhabiting the Korean Peninsula.

Ultimately, it is important to study the biogeography of pseudocrangonyctids in order to better understand the origin and evolution of subterranean amphipod fauna in the Far East (Sidorov & Holsinger, 2007). Further molecular phylogenetic analyses of Pseudocrangonyx are essential for enhancing the understanding of subterranean Crangonyctoidea species diversity and evolutionary history in Far East Asia.

Conclusions

This study described three new species described found in caves in Korea. Two new species of them were found from caves treated as the morphological variants of P. asiaticus Uéno, 1934. The other new species was found from a cave with no former records of the genus Pseudocrangonyx. These new species may receive a unique species status within the genus Pseudocrangonyx based on our morphological examination and molecular analyses. These results suggest that the genus Pseudocrangonyx may have greater species diversity in the Korean Peninsula than previously believed. Although we failed to obtain molecular data of the originally described P. asiaticus, obtaining those data in future studies may make it possible to determine the true species diversity of the subterranean amphipod Pseudocrangonyx in Far East Asia including the Korean Peninsula.

Supplemental Information

Supplemental Information 1 28S of three new species.

Click here for additional data file.

Supplemental Information 2 COI of three new species.

Click here for additional data file.

The authors are grateful to Dr Christopher J Glasby, Dr Charles Oliver Coleman, and an anonymous reviewer for their constructive comments on this manuscript. We are grateful to the Korean Society of Cave Environmental Science (Daejeon, Korea), especially Mr. Yong Gun Choi for their continued support and assistance. We are grateful to the Cultural Heritage Administration and the National Institute of Ecology, Korea for supporting the cave research. The first author is very grateful to Dr Ko Tomikawa (Hiroshima University, Japan) for the help provided.

Additional Information and Declarations

Competing Interests

Author Contributions

DNA Deposition

Data Availability

New Species Registration

The authors declare there are no competing interests.

Chi-Woo Lee conceived and designed the experiments, performed the experiments, analyzed the data, prepared figures and/or tables, authored or reviewed drafts of the paper, and approved the final draft.

Gi-Sik Min conceived and designed the experiments, authored or reviewed drafts of the paper, and approved the final draft.

The following information was supplied regarding the deposition of DNA sequences:

The partial sequences of 28S rRNA gene are available at GenBank: MW026427, MW026433 and MW026439.

The partial sequences of COI gene are available at GenBank: MW026424 to MW026426, MW026430 to MW026432 and MW026436 to MW026438.

The following information was supplied regarding data availability:

The genetic information newly obtained in this study are available in the Supplementary Files.

The following information was supplied regarding the registration of a newly described species:

Publication LSID: urn:lsid:zoobank.org:pub:A60F095A-2A50-4D87-876C-6D8E3D8539CE

Pseudocrangonyx deureunensis Lee & Min sp. nov. LSID: urn:lsid:zoobank.org:act: 462F3F22-7C4B-4B24-9826-A752A770C897

Pseudocrangonyx kwangcheonseonensis Lee & Min sp. nov. LSID: urn:lsid:zoobank.org:act:258FD1A7-1A6E-46BB-B5BA-E5C33AC708B4

Pseudocrangonyx hwanseonensis Lee & Min sp. nov. LSID: urn:lsid:zoobank.org:act:C4DDF1EC-47C1-4B6D-BDEC-95CE277798F3.

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
