# Peer review of "Three new species of subterranean amphipods (Pseudocrangonyctidae: Pseudocrangonyx) from limestone caves in South Korea"

_PeerJ, doi:10.7717/peerj.10786_

## Round 0.1 · original submission · Minor Revisions

Dear Author, please find attached the reviews of two experts in your field, and my editorial comments. I concur with the reviewers that the ms requires only minor changes to be acceptable for publication. Please address all suggestions and return a revised version. Best, Chris

n

·

Basic reporting

This is a very nice taxonomic paper and it had been a pleasure to review it! I wished all taxonomic papers that I reviewed are so well prepared! It has been written very carefully. There a very few language problems. I tried to adress these directly in the text, but the authors must be aware, that I am not a native speaker (despite my name!).

The citation of the literature is well done and the important papers related to the question are listed. The references are well formatted... nothing to complain about.

The structure of the article is very well balanced.

The illustrations are very nice and clear. The taxonomic descriptions are very detailed, the results are well presented and the hypotheses well supported.

Experimental design

The methods used fit the research questions and are standard for modern taxonomy. No complains!

Validity of the findings

The results are novel and very interesting. They form the basis for many future studies in different fields (evolution, ecology ...) on this groundwater taxon.

Additional comments

My comments are in the uploaded pdf. Here are some general comments:
1) At first appearance of a species in the text the genus name should be spelled out and author and year should be added.
2) Suggestion for your next paper: Gum chloral as an embedding medium for microscopic slides is not very long-lasting. From the conservation point of view they are a desaster after some decades. Then they often crystallize and leaving the slides useless. Better use Canada Balsam or Euparal.
3) Suggestion for your next paper: It appears to me that for the drawing a bitmap drawing program had been used, as some of the details have a higher line-weight. With vector graphics this could be avoided, then all lines can be scaled and keep their weight. Not a big problem, though, but the plates could look a bit better then.
4) In the manuscript you are referring to the "true P. asiaticus" several times. I know what you mean, but could you perhaps use "the orginally described P. asiaticus" or "P. asiaticus sensu stricto" (or s.str.) and explain this in Material and Methods?
Thanks, dear authors! Best regards, Ch. Oliver Coleman (Museum für Naturkunde Berlin, Germany)

Reviewer 2 ·

Basic reporting

no comment

Experimental design

no comment

Validity of the findings

no comment

Additional comments

“Three new species of subterranean amphipod (Pseudocrangonyctidae: Pseudocrangonyx) from limestone caves in South Korea”

The manuscript described three new species of the subterranean Pseudocrangonyx, and provided a molecular phylogeny to determine the relationship of Korean cave species. The description and illustrations are excellent. The phylogenetic analyses are reasonable. I recommend to publish it with minor revion.


1) Line 128: urosomite 1–3 should be urosomites 1–3
2) There are 6 species from Korea in Table 1, however, only 5 species are used for genetic distance in Table 2. Please explain.
3) For figure legends, “ventral view”, “posterior view”, “medial view” are different?

---

## Round 0.2 · accepted · Accept

Dear Author, thank you for considering, and incorporating, the suggestions of the reviewers. (also for seeing to the English language editing). The paper looks very nice! best Chris